# Remote Sensing, Geophysics, and Modeling to Support Precision Agriculture—Part 1: Soil Applications

**Arya Pradipta** [1,*], **Pantelis Soupios** [1], **Nektarios Kourgialas** [2,*], **Maria Doula** [3], **Zoi Dokou** [4], **Mohammad Makkawi** [1], **Mohammed Alfarhan** [5], **Bassam Tawabini** [1], **Panagiotis Kirmizakis** [1] **and Mohamed Yassin** [6]

1   Department of Geosciences, College of Petroleum Engineering & Geosciences, King Fahd University of Petroleum & Minerals, Dhahran 31261, Saudi Arabia; panteleimon.soupios@kfupm.edu.sa (P.S.); makkawi@kfupm.edu.sa (M.M.); bassamst@kfupm.edu.sa (B.T.); p.kirmizakis@kfupm.edu.sa (P.K.)
2   Institute for Olive Tree Subtropical Crops and Viticulture, Hellenic Agricultural Organization (H.A.O.-DEMETER), 73134 Chania, Greece
3   Laboratory of Non-Parasitic Diseases, Benaki Phytopathological Institute, 14561 Athens, Greece; mdoula@otenet.gr
4   Department of Civil and Environmental Engineering, California State University, Sacramento, CA 90032, USA; zoi.dokou@csus.edu
5   Remote Sensing Lab, College of Petroleum Engineering & Geosciences, King Fahd University of Petroleum & Minerals, Dhahran 31261, Saudi Arabia; mohammed.alfarhan@kfupm.edu.sa
6   Interdisciplinary Research Center for Membranes and Water Security, King Fahd University of Petroleum and Minerals, Dhahran 31261, Saudi Arabia; mohamedgadir@kfupm.edu.sa
*   Correspondence: arprdipta@gmail.com (A.P.); kourgialas@elgo.iosv.gr (N.K.)

**Abstract:** Sustainable agriculture management typically requires detailed characterization of physical, chemical, and biological aspects of soil properties. These properties are essential for agriculture and should be determined before any decision for crop type selection and cultivation practices. Moreover, the implementation of soil characterization at the beginning could avoid unsustainable soil management that might lead to gradual soil degradation. This is the only way to develop appropriate agricultural practices that will ensure the necessary soil treatment in an accurate and targeted way. Remote sensing and geophysical surveys have great opportunities to characterize agronomic soil attributes non-invasively and efficiently from point to field scale. Remote sensing can provide information about the soil surface (or even a few centimeters below), while near-surface geophysics can characterize the subsoil. Results from the methods mentioned above can be used as an input model for soil and/or soil/water interaction modeling. The soil modeling can offer a better explanation of complex physicochemical processes in the vadose zone. Considering their potential to support sustainable agriculture in the future, this paper aims to explore different methods and approaches, such as the applications of remote sensing, geophysics, and modeling in soil studies.

**Keywords:** soil properties; precision agriculture; sustainable development; remote sensing; agricultural geophysics; numerical approaches

## 1. Introduction

The future projection of population growth is expected to increase the food demand, with implications of a massive expansion of global agriculture areas. Srinivasan et al. [1] predicted that the present rate of increasing agricultural yield would not satisfy the projected rising food demand in 2050 and beyond; thus, new techniques and innovations are necessary to overcome this challenge. Furthermore, since food production is highly linked to soil, without proper agricultural management, the environment and natural resources' sustainability will be threatened as well.

Nowadays, the total land area is estimated at around $130{,}575{,}894$ km$^2$. However, only around 12% is suitable for crop production without many limitations, while only 3% of the

total land area is considered highly productive [2,3]. This is because not all agricultural soils are fertile and productive. Moreover, and regarding human interventions, not all soils are used efficiently. With few exceptions, the times from antiquity to the present day are characterized by the absence of a strategic demarcation of soil use-zones according to soil's physical and chemical properties and in combination with local climatic conditions. This led to soil over-exploitation without a plan and without considering its future sustainability. Therefore, employing tired and degraded soil to continue providing food for the growing population of the planet will be challenging.

Soil quality protection is part of sustainable soil management and becoming a necessity globally. In the framework of agriculture, the meaning of soil quality is defined as the ability of soil to perform its function of sustainable agricultural production and enable it to respond to sustainable land management [4]. Sustainable soil management also includes precision agriculture, which refers to the successful implementation of the identification and understanding of important parameters in order to design the appropriate management plans successfully. One of these parameters is the detailed understanding of soil physicochemical properties that are important for agricultural practice.

The significance of precision agriculture has gained the widespread attention of scientific communities as it has become a critical issue amid the increasing worldwide food demand. Soil properties such as soil water content, organic matter, soil nutrient, soil texture, and soil structure are some typical agronomic parameters regularly monitored by farmers. Traditionally, these properties are assessed by in situ measurement, sampling, and followed by laboratory analysis. These methods are invasive, laborious, time-consuming, and cannot represent a larger area. In this context, spatial and timely observations are crucial to capture the variability of soil properties. The development of soil sensors, ranging from electrical, electromagnetic, optical, and radiometric to mechanical sensors, offers opportunities to improve the effectiveness of soil monitoring [5].

Currently, non-invasive techniques such as remote sensing, geophysics, and soil modeling have been successfully employed in agricultural studies by a number of researchers. The aforementioned techniques provide valuable means for the characterization of soil properties. Physicochemical information required by farmers can be retrieved through remote sensing and geophysics techniques, while modeling can increase our understanding of complex soil processes. Therefore, this paper intends to highlight remote sensing, geophysics, and numerical soil modeling techniques in soil studies along with their applications to support precision agriculture.

## 2. Agronomic Soil Properties

The first step for implementing precision agriculture is the detailed characterization of the local conditions in terms of the substrate to be treated (i.e., soil) as well as the factors that impact it. Deep knowledge of the processes of the system to be cultivated and not merely the chemical composition of the soil is very crucial. When designing a precision agricultural system, it is necessary to know the soils' properties, as well as their spatial distribution. This is the only way to develop appropriate interventions and cultivation practices that will ensure the required nutrient supply in an accurate and targeted way and ensure compliance with sustainability principles.

The main soil properties that should be known before any decision for crop type selection and cultivation practices are physical, chemical, and biological. For fertilization purposes, the available nutrients, such as the exchangeable forms of cations (i.e., potassium, magnesium, calcium), need to be known instead of their total soil concentration. This is a crucial point for understanding soil constituents. The physical properties include soil texture (e.g., the presence of clay, sand, and silt) and soil structure, while the chemical properties vary from nutrients to organic matter [6,7]. Biological properties are often overlooked or ignored, despite their particular importance in maintaining the quality of a soil system. Ideally, biological properties should also be considered to protect and enhance biodiversity, which will ensure yield and soil health [8].

Soil texture is one of the most important physical soil properties since it interacts with water and chemical elements. In general, this soil attribute is determined through particle size analysis. Fine-textured soils are more fertile than coarse-textured soils because they contain a higher amount of organic matter and exchangeable cations. Fine-textured soils are also richer in nitrogen, available nutrients, and microbial population than coarse-textured soils. On the other hand, sandy soils lose nutrients more easily, and are wetted with lesser amounts of water, but they quickly dry up, making frequent and well-scheduled irrigation a critical aspect of the management plan [2].

Many processes in the soil layer are governed by soil structure, such as water storage and infiltration, nutrient recycling, gaseous exchange, root development, erosion susceptibility, and other physical supports [9]. Poor soil structure would bring adverse consequences for agricultural productivity. For example, reduced soil permeability due to soil compaction could affect root development, nutrient uptake, and the amount of fertilizer used for crop production, and restrict the irrigation water infiltration, which might result in runoff and soil erosion, as well as increased sodium levels [10–13]. It can be concluded that a well-developed soil structure has a critical role in a functioning and productive soil [2,14]. Soil texture and structure together regulate the porosity, density, compactness, retention, and movement of water and air in soil, with the latter having a crucial role in the respiration of plant roots and microorganisms and the transformation of minerals and organic matter [15].

The availability of nutrients is influenced directly or indirectly by factors such as the type of organic matter and its decomposition extent, soil content in colloidal parts, water content, soil structure, climatic conditions, and many other parameters, for which there is no general guide to assist farmers in predicting nutrient availability. The understanding of nutrient availability is important for farm management as it acts as a basis to estimate nutrient use efficiency, maintain nutrient stock, and avoid groundwater contamination. Limited freshwater and land availability for crop production, high inorganic fertilizer costs, and increasing concerns of environmental pollution make the prediction of nutrient efficiency and accuracy a key factor to support precision agriculture [16,17].

Soil organic matter (SOM) represents soil fertility and is regarded as an essential factor governing the dynamics of agrochemicals in the soil [18]. In particular, the spatiotemporal variabilities of soil organic matter (SOM) are related to land use, climate, soil type, and agricultural management [19]. Since SOM is essential for soil health, its variabilities should be assessed to achieve sustainable soil management across various scales. Higher SOM concentrations are associated with water storage capacity, regulation of nutrients, and soil aggregate stabilization, which lead to soil structure improvement [20]. These factors are crucial to enhancing irrigation water management and crop yield [20]. This organic compound can be derived from living organisms or atmospheric carbon dioxide [21]. The erosion process reduces the concentration of SOM, resulting in lower productivity. This condition can be avoided through spatial information of SOM; thus, the chemical fertilizer can be applied precisely to degraded soil.

Another essential physical soil attribute is soil moisture (SM), representing the amount of water stored in the non-saturated soil zone. Adequate information of SM is critical for agricultural water management to improve crop water availability, irrigation water efficiency, and water productivity, and to reduce the potential environmental impact [22]. The role of SM is not limited to supporting vegetation growth, but also governing rainfall partition into infiltration and runoff and regulating soil evaporation and transpiration [23]. The latter role of SM has other impacts on climate processes such as precipitation and air temperature [24].

Soil salinity and compactions are degrading factors representing the major problems in modern agriculture. These factors should be considered since they can bring adverse effects not only to plant growth and crop yield but also to the economy. Salinity typically occurs in arid and semiarid areas, where evapotranspiration exceeds precipitation and when the continuous use of irrigation water contains an adequate amount of dissolved

salt [25]. This degrading process can be exaggerated by the expansion of farmland with insufficient natural water resources [26]. Meanwhile, soil compaction in agriculture areas is generally accelerated by anthropogenic activities, such as intensive tillage and heavy farm machinery, crop rotation, intensive grazing, and unsustainable soil management [27,28]. Soil compaction can be defined as densification processes to decrease soil porosity and permeability [29]. This process can cause a loss of habitat quality of microfauna and crops, reducing hydraulic conductivity and increasing the nutrient demand [12].

## 3. Remote Sensing Observations

Rapid developments of remote sensing technology in the last few years offer a viable option in mapping soil properties at a farm scale to support real-time management. Their spatial resolutions, revisit time, and spectral resolutions have been improved tremendously. Diverse imagery data can be obtained from spaceborne, airborne, and unmanned aerial vehicles (UAVs). Based on the spectrum domain, remote sensing can be classified into an optical, thermal, passive microwave, and active microwave. The choice of remote sensing platform and spectrum domain usually depends on specific applications (i.e., SM, soil texture, salinity) and the size of the field. UAVs could offer low-cost alternatives for agricultural monitoring among remote sensing platforms, especially for small farms where the resolution is suitable enough to observe the soil properties variabilities. An overview of remote sensing technologies and their applications in soil studies is shown in Table 1.

### 3.1. Optical Remote Sensing

Visible (VIS), near-infrared (NIR), and shortwave infrared (SWIR) reflectance spectroscopy is considered a cost-effective and time-efficient tool for soil property characterization [30–32]. Besides laboratory- and field-based spectroscopy, the feasibility of soil property assessment through airborne and spaceborne platforms has been highlighted by many studies. The principle of the spectroscopy method is based on the interaction between electromagnetic radiation and matter. Due to the presence of physical and chemical characteristics in the soil, such as minerals, organic matter, and water molecules, the electromagnetic radiation will cause individual molecular bonds to vibrate, either by bending or stretching [33–35]. These vibrations will lead to the absorption of light with a specific energy quantum that is related to frequency, allowing for soil property analysis [33].

In particular, the spectral region of VIS, NIR, and SWIR ranges from 400 to 700 nm, 700 to 1100 nm, and 1100 to 2500 nm, respectively [36]. These spectral regions have been found to be sensitive to several soil properties, such as SM, SOM, soil texture, soil nutrients, and salinity. The reflectance spectra are mostly non-specific, broad, and overlapping bands; hence, multivariate statistics are required to correlate spectra with soil properties [37]. These methods include partial least square regression (PLSR), principal component regression (PCR), and stepwise multiple linear regression (SMLR), which are frequently used [38,39]. Besides these, other techniques have also been used—for example, artificial neural network (ANN) [40], regression tree (RT) [41], multivariate adaptive regression splines (MARS) [34], and support vector machines (SVM) [42]. The use of machine learning algorithms is increasing as the relationships between spectral reflectance and soil properties are not always linear [43]. Pre-processing transformation based on mathematical functions is also commonly applied to correct nonlinearities, measurement and sample variations, and noisy spectra [44].

Soil color has been used as an indicator for SOM and SM monitoring through remote sensing [45–47]. For instance, a higher SOM concentration generally is characterized by a darker color; therefore, the spectral reflectance will be low and vice versa [32]. Furthermore, SOM measurement typically utilizes VIS–NIR–SWIR spectral regions [37,48,49]. To date, there is still no consensus about the most indicative wavelength for SOM assessment [50]. Previous studies reported that the wavelength related to SOM varies in the region of 820 to 2300 nm [51–53]. Usually, the accuracy of spectroscopy in SOM retrieval is low in soil that contains low organic matter and a high sand fraction [18,44,54]. Francos et al. [50]

found that the accuracy of SOM measurement with organic matter less than 0.6% tends to decrease. One of the challenges in SOM assessment is that SM has a strong effect on the spectral signal. In order to improve SOM measurement, previous studies used the normalized soil mixture index (NMSI) to quantify the effect of SM [37,49].

The quantification of SM through spectroscopy generally employs a longer wavelength such as SWIR since the spectral reflectance in this region has better sensitivity to SM compared to VIS and NIR [55,56]. Many studies reported that spectral reflectance at 1400, 1900, and 2200 nm can be used as an indicator to assess SM [56–58] Compared to SWIR, the spectral reflectance at the NIR domain is more sensitive to soil characterized by large reflective particles and a large pore space, such as quartz sand [58]. Therefore, the use of the NIR domain might not be applicable in agricultural soils that are dominated by opaque particles and a wider size distribution.

The ability of reflectance spectroscopy through space-based remote sensing to predict soil elements, particularly soil nutrients, has been demonstrated by several studies [59,60]. Total nitrogen [17,61–63], phosphorus [17,62], and potassium [17,62] from different soil samples have been successfully predicted using VIR and NIR sensors and various multivariate calibration methods. Nitrogen, phosphorus, and potassium are reported by a previous study to have small absorption reflectance in the SWIR region at 1900 to 2370 nm, 2360 to 2380 nm, and 1900 nm to 2280 nm, respectively [62]. Among them, only nitrogen has good accuracy with a higher coefficient of determination. Other studies showed that nitrogen, phosphorus, and potassium also have adsorption features in the VIS and NIR regions [60,64]. In particular, the key wavelength to predict potassium is reported to be in the VIS region [64].

Sand, silt, and clay are typical soil texture variables retrieved from remotely sensed images. A previous study showed that soil reflectance in the spectral region was correlated with soil texture [65]. Soil texture compositions are assessed based on a portion of the electromagnetic spectrum reflected by soil components [66]. In general, higher spectral reflectance is associated with sandy soil due to the presence of quartz, while clay content presents low reflectance and a flatter curve [67]. This is because quartz is known as a mineral with low absorption that leads to higher reflectance. Clayey soil might lead to underestimation, while very sandy soil could generate overestimated measurements [67]. In particular, the spectral wavelength of 2200 to 2300 nm is considered significant for clay mineralogy and therefore can be utilized to retrieve the spectral response of clay directly [67,68].

In salt-affected soil, the presence of salts will affect the absorption bands [69]. For instance, gypsum has strong adsorption in the region of 1000 to 2500 nm, while halite (NaCl) has adsorption features in the region of 980 to 1950 nm [70]. The use of SWIR in soil salinity characterization is more suitable since the spectral reflectance in the VIS–NIR regions might show severe confusion due to the effect of soil properties [70].

### 3.2. Microwave Remote Sensing

The general advantage of microwave remote sensing is that it is less affected by atmospheric conditions due to the low microwave frequency used for Earth surface monitoring; thus, the observation can be made at any time [71]. In essence, there are two basic approaches in the utilization of microwave remote sensing: active and passive sensors. The active sensor emits and receives a microwave pulse, while the passive sensor records the natural thermal emission from the land surface [72]. In an active sensor, the power of the received signal is compared with the emitted signal to determine the backscattering coefficient, which is related to the characteristics of the object's surface [72]. Microwave remote sensing can be affected by numerous factors, such as the dielectric constant, backscattering coefficient, surface roughness, bulk density, soil texture, vegetation cover, incident angle, bands, and polarization [73]. The wavelength of the electromagnetic spectrum ranges from 1 to 100 cm and is divided into several bands referring to letter systems, such as X (0.8 cm), K (3 cm), C (5 cm), S (10 cm), L (20 cm), and P (50 cm) [72].

It has been widely recognized that microwave remote sensing is the most viable technique in terms of SM monitoring. The basic principle of SM measurement through microwave sensing is the large dielectric contrast between liquid water and dry soil [71,74,75]. In general, SM retrieval is based on the radiative transfer model and dielectric mixing model. The L-band frequency has an advantage in SM assessment since it has the ability to penetrate into the subsoil more deeply than other frequencies. Subsequently, the L-band frequency could provide not only SM information on the top of the soil surface but also at the depth of up to a few centimeters. A series of satellites have been developed and utilized for soil global moisture observation, involving passive and active microwave sensing. Among them, active microwave sensing with the synthetic aperture radar (SAR) configuration can meet the spatial resolution required for soil management at a farm scale [76].

The possibility of SOM retrieval through a microwave remote sensing system is highlighted by Jonard et al. [77] and Bircher et al. [78]. The SOM estimation is based on the concept that organic matter will generate macropores, resulting in high porosities, low bulk densities, high water-holding capacities, and high potential water infiltrations, thus allowing SOM to be calculated through the same algorithm used for SM retrieval [77,78].

The change in the backscattered microwave signal due to soil texture is small; thus, it might not be possible to retrieve soil texture components directly from the signal [79]. This is probably why the studies of microwave sensing applications for soil texture analysis are still limited. However, several studies have still explored the feasibility of SAR to assess soil texture [66,80,81]. In general, the variation of soil texture has a linear relationship with SM over time. For instance, SM will increase with increasing clay (slow drying), while an increasing sand fraction (fast drying) corresponds to decreasing SM [66,79].

The measurement of soil salinity through microwave remote sensing is based on soil dielectric properties since salinity is associated with electrical conductivity [47]. Microwave sensing can be considered an efficient tool in monitoring saline soil due to the differential behavior between the real and imaginary parts of the dielectric constant, which allows the separation of salt-affected soil from others [69]. The real part is independent of soil salinity and alkalinity, while the imaginary part is sensitive to soil electrical conductivity variabilities [69], thus affecting the backscattering coefficient [82]. Despite being able to be performed under cloudy conditions, the application of this sensor in soil salinity monitoring is still rare.

### 3.3. Thermal Remote Sensing

Thermal remote sensing works by measuring the emitted radiation from the object on the surface and converting it into temperature [83,84]. Every object on the surface with a temperature above 0 K or 273 °C will emit radiation in the thermal infrared (TIR) region of the EM spectrum [85]. The sensor in thermal remote sensing can detect thermal radiative properties from objects as they are more intense than the reflected solar radiation [85]. Moreover, the higher the temperature of the object, the greater the emitted radiation will be [83].

Compared to optical and microwave remote sensing, the applications of thermal remote sensing in soil studies are still limited to SM and soil texture observation [83]. The most common method used for SM retrieval is the triangle method, which employs the vegetation index and surface temperature. The vegetation index is calculated from optical imagery, while the surface temperature is derived from a thermal sensor. Generally, the decrease in SM is followed by a decrease in the vegetation index and increase in surface temperature [86]. The determination of soil texture is essentially based on the linear relationship between the surface temperature retrieved by thermal remote sensing, SM variations, and soil texture [87]. For example, sand-dominated soil is expected to have low water content during the dry period, leading to a higher surface temperature [87].

**Table 1.** The highlight of remote sensing groups and their applications in agricultural soil studies.

| Groups | Soil Attributes | Spectral Ranges | Platforms | References |
|---|---|---|---|---|
| Optical | Soil organic matter | VIS–NIR–SWIR | Sentinel-2 | [18] |
| | Soil moisture | NIR–SWIR | Landsat-8 | [88] |
| | Soil nutrient | VIS–NIR | HuanJing-1A | [60] |
| | | VIS–NIR–SWIR | Earth Observing-1 | [59] |
| | Soil texture | VIS–NIR–SWIR | Sentinel-2 | [18] |
| | Soil salinity | VIS–NIR–SWIR | Landsat 8 | [89] |
| | | VIS–NIR–SWIR | Sentinel-2 | [70] |
| Microwave | Soil organic matter | L-band | ELBARA-II | [73] |
| | Soil moisture | X-band | AMSR-E | [90,91] |
| | | | AMSR2 | [90–92] |
| | | | FY3B | [91] |
| | | C-band | AMSR2 | [93] |
| | | | ASCAT | [94] |
| | | | Sentinel-1 | [95,96] |
| | | L-band | SMOS | [90,92,94,97] |
| | | | SMAP | [90,92,98] |
| | Soil texture | X-band | TerraSAR-X | [79,81] |
| | | C-band | Sentinel | [66] |
| | Soil salinity | C-band | Radarsat-2 | [99] |
| | | L-band | PALSAR | [99] |
| Thermal | Soil moisture | TIR | MODIS | [100] |
| | Soil texture | TIR | MODIS | [87] |

## 4. Geophysical Survey

Along with remote sensing technology, the use of geophysical methods will become an essential tool to support sustainable agriculture in the future. Due to its non-invasive technique, the near-surface geophysical survey has been widely utilized to improve soil management and investigate the soil properties. Geophysics can provide high-resolution, 2D, 3D, or time-lapse (4D) mapping of the soil, tackling the heterogeneity and complex dynamics of a soil system. Specifically, electrical resistivity imaging (ERI), electromagnetic induction (EMI), and ground-penetrating radar (GPR) are the most common geophysical methods used for agricultural applications. Soil properties and variables, such as porosity, density, clay content, soil moisture, and salinity, are the typical parameters measured and extracted through a geophysical survey. Besides these, magnetometry, self-potential, and seismic are three promising geophysical methods that can be complementarily applied for the same purposes. This section will highlight some geophysical techniques commonly used and their applications to support precision agriculture. Table 2 shows a brief overview of geophysical applications in soil studies.

### 4.1. Electrical Resistivity

Electrical resistivity (ER) is one of the most employed geophysical techniques for soil studies. The general objective of ER acquisition is to map the resistivity distribution within the soil volume since ER is a function of the solid constituent, arrangement of the void, saturated water, fluid, and temperature [101]. This method can be performed by injecting continuous electric currents into the subsurface through the deployed current

electrodes and measuring the potential difference between the potential electrodes, trying to reconstruct the subsoil heterogeneities [101,102]. In particular, ER measures the bulk soil resistivity and reciprocal of the apparent soil electrical conductivity and requires good contact between soil and electrodes [33]. Electrical Resistivity Tomography (ERT) is an extension of ERI that is capable of providing 2D, 3D, and 4D imaging of the surface and subsurface. In this method, low-frequency electrical currents are injected from a series of electrodes placed on the ground surface and subsurface, thus allowing the user to measure the potential distribution.

Among the various soil physical properties, water content, bulk density, and clay content are important parameters that would affect the soil electrical resistivity (ER) [103]. Previous studies have already shown the applicability of ER measurements to investigate the SM [104,105], soil layers [106], and soil degradation processes such as compaction [27,103,107,108] and salinity [109]. An increase in the water content is expected to result in low resistivity. Similarly, a rich organic soil with dissolved materials/ions into the flowing groundwater makes the site electrically conductive. Tabbagh et al. [106] highlighted the use of the electrical resistivity method to delineate the clay layer in a sandy environment that might affect the hydrological characteristics of the subsurface. This information could help farmers or decision-makers to determine the soil quality from the context of precision agriculture and ensure that the optimum irrigation plan is spatially applied and the proper quantity of fertilizer is employed in different parts of the field. On the other hand, even though ER is sensitive to bulk density, it cannot be performed as a direct method to quantify the degree of soil compaction [107]. The quantitative relationship between electrical conductivity, water content, and compaction should be established to achieve the full potential of ER for soil compaction measurement [103].

*4.2. Electromagnetic Induction*

Electromagnetic induction (EMI) is a non-invasive method that is based on Faraday's law in physics and is mainly deployed to detect conductive material in the subsurface [110]. EMI consists of transmitter and receiver coils installed on both ends of nonconductive bars [33]. The transmitter injects a primary electromagnetic field, inducing electrical currents that later generate a secondary magnetic field in the soil [111]. The magnitude and phase of the secondary magnetic field read by the receiver differ from the primary magnetic field due to soil properties, and this is used to measure the apparent soil electrical conductivity (ECa) [33,111]. The EMI's depth of investigation (DOI) varies with coil spacing and frequency [33,110]. For example, an increase in frequency will lead to a decreasing DOI, while increasing the coil spacing will lead to an increasing DOI [33,112]. In many studies, the vertical resolution of the soil profile obtained by EMI is low; therefore, the use of a new EMI instrument with multiple coil separations and orientations is recommended to improve the resolution [113]. Moreover, the use of multiple coils using multiple frequencies will enable simultaneous measurement at various depths [114].

EMI has repeatedly been used to map soil characteristics among geophysical techniques due to its fast measurement and sensitivity since the induction principle does not require direct contact with the ground surface [115–117]. EMI is initially utilized for soil salinity mapping [111]. In saline conditions, variations in ECa measured by EMI are derived from the concentration of soluble salt, which is regarded as the dominant physicochemical factor affecting conductivity [111]. Other factors affecting EMI measurement are the saturation percentage, bulk density, or temperature [118]. In order to transform the EMI measurement into soil salinity, a salinity probe is commonly used, as a ground-truthing. In practice, EMI can be combined with other methods, such as magnetic and radiometric methods, to assess soil body information that might help to identify the spreading of soil salinity [115].

Under non-saline conditions, ECa is mainly associated with several soil properties. The applications of EMI have been widely demonstrated to monitor SM [119–122], SOM [123], and soil texture [124,125]. Overall, increasing soluble salt concentration, soil water, and

clay content will lead to increasing apparent electrical conductivity [111]. Among these soil properties, EMI has a poor correlation with soil texture compared to SM and SOM [126]. In practice, the combination of several soil properties (e.g., soil salinity, soil texture, soil moisture) could lead to misinterpretation of the dominant parameters affecting ECa. For instance, high ECa could be associated with high soil salinity or clay content [124]. Several techniques have been proposed to determine the predominant factor influencing ECa, including simple statistical correlation and wavelet analysis, where the latter approach is considered the more powerful method [118].

### 4.3. Ground-Penetrating Radar

In principle, ground-penetrating radar (GPR) transmits high-frequency electromagnetic (EM) waves into the subsoil and records the scattered, reflected, and refracted waves in a non-invasive manner [127]. The interaction of the electromagnetic waves at the air–soil interface would allow the user to estimate the dielectric permittivity and electrical conductivity [128,129]. The dielectric permittivity is related to the EM wave velocity, while the electrical conductivity is associated with the EM wave attenuation [130]. These variables vary depending on the soil physical properties, thus allowing the opportunity to characterize soil properties.

GPR can image subsoil in 2D and 3D with high spatial resolution up to the depth of several meters, particularly in low-conductivity media [33]. On the other hand, the resolution and range of GPR will decrease with the presence of clay or electrically conductive soil [127]. There are three different GPR configurations that are widely employed, involving cross-borehole, airborne, and surface [131]. Since this method relies on the reflector at a known depth, determining the origin and depth of the reflector should be considered to provide accurate subsurface interpretation [132].

Prior studies highlighted successful GPR applications in precision agriculture—for example, SM [132–134], SOM [135,136], soil texture [137], soil horizons [138], and soil compaction [139–141]. SM investigation is the most frequent GPR application in soil studies due to GPR's ability to measure permittivity, which is strongly related to soil water content. Several approaches have been proposed to determine SM using GPR, including signal velocity analysis, reflection amplitude analysis, early time signal analysis, and full-waveform analysis [142]. Another application of GPR to support precision agriculture is SOM layer mapping. Separation of SOM and non-SOM layers is possible due to the significant dielectric constant change between them, even though the distinction of each SOM component remains a challenge [136].

### 4.4. Seismic

The seismic method works by measuring the propagation velocities of seismic waves through the subsurface from the source to the receiver, called a geophone. Time arrivals and amplitudes of direct, refracted, and reflected seismic waves recorded by geophones will be used to characterize the subsurface [143]. Although not as popular in soil monitoring as other geophysical methods, the seismic geophysical method still can provide valuable information on soil compaction. The potential of this method to observe soil compaction has already been demonstrated by Donohue et al. [144], Keller et al. [145], and Romero-Ruiz et al. [14]. Since soil compaction is related to bulk density, seismic wave velocity would also increase [145]. Keller et al. [145] exhibited that microseismic measurement based on P-wave velocities can be used to monitor the dynamics of soil behavior due to loading with an agricultural tire. In addition, the use of surface waves was also explored by Donohue et al. [144] for soil compaction assessment. The result showed that their approach had a favorable correlation with on-site geotechnical measurement.

**Table 2.** The geophysical methods and their applications in agricultural soil studies.

| Geophysical Methods | Soil Properties | Applications | References |
|---|---|---|---|
| Electrical Resistivity | Soil moisture | Soil moisture variations | [104,146,147] |
| | | Identifying root water uptake | [148,149] |
| | Soil structure | Soil-bed rock delineation | [150] |
| | | Identification of compacted zones | [102] |
| | | Characterization of regolith | [151] |
| | | Soil structural change after compaction | [141,152] |
| Electromagnetic Induction | Soil moisture | Soil moisture variations | [114,121,153] |
| | Soil texture | Identification of clay, silt, and sand/gravel | [124,154,155] |
| | Soil organic matter | Soil organic matter mapping | [123,156] |
| | Soil salinity | Soil salinity distribution | [116,157] |
| | Soil structure | Detection of soil compaction | [158] |
| Ground-Penetrating Radar | Soil moisture | Soil moisture measurement | [132,134] |
| | Soil texture | Spatial variations of clay content | [155] |
| | Soil structure | Identifying the compacted layer | [139,159] |
| | | Delineation of soil and bed rock | [160] |
| | Soil organic matter | Identifying humous and non-humous layers | [136] |
| Seismic | Soil structure | Detection of compacted soil | [14,144] |

Although all the above information makes geophysical methods a well-suited tool for precise agriculture management, they cannot completely replace conventional substrate characterization and sampling approaches, such as the geochemical characterization of soil at specific points at different depths, which can be used to constrain the geophysical uncertainty. Overall, the complementary use of geophysics with remote sensing and geochemistry can improve significantly the soil characterization for agricultural purposes.

## 5. Soil Modeling

Soil processes are typically nonlinear and governed by time-variable boundary conditions requiring numerical approaches to determine soil states and fluxes. Originally, soil modeling aimed to be related to agricultural applications to optimize soil conditions and, subsequently, crop productivity. Since soil modeling is a complex process, the input data requirements would be more demanding [26]. The input data of soil modeling typically range from meteorological conditions (e.g., rainfall, temperature, humidity, radiation) and phenological characteristics (e.g., LAI, root depth, albedo, canopy roughness), to hydro physical properties (e.g., soil–water retention, soil hydraulic conductivity) [161]. In addition, remotely sensed imageries and proximal data sensing such as geophysical acquisition could provide the required data input to drive a model. To date, Richards' equation for water flow and the convection–dispersion equation for solute transport are the most common techniques used in soil modeling [162]. Hence, these two equations will be discussed in this section. Table 3 shows different numerical models solving Richards' and the convection–dispersion equation and their applications.

### 5.1. Richards' Equation for Water Flow

Hydrological fluxes in the unsaturated porous media are considered the most complex natural process and crucial for agricultural management. Commonly, semi-empirical, empirical, and physical-based models are employed to analyze hydrological fluxes in the vadose zone, such as infiltration [163,164]. The semi-empirical model uses simple forms of the continuity equation. Empirical models are derived from field or laboratory analysis, while physical-based models are based on the law of conservative mass and Darcy law [164]. Among them, the physical-based model that involves Richards' equation offers the capability to provide a detailed explanation of water fluxes within the soil profile [163]. The Richards equation can be described below:

$$\frac{\partial \theta(h_w)}{\partial t} = \frac{\partial}{\partial z}\left[k(h_w)\left(\frac{\partial h_w}{\partial z} + 1\right)\right] - S(z) \tag{1}$$

where $z$ is the vertical coordinate positive upward (L), $t$ is time (T), $h_w$ is the soil–water pressure head (L), $\theta$ is the volumetric water content ($L^3 L^{-3}$), $S(z)$ is a sink term representing water uptake by plant roots ($L^3 L^{-3} T^{-1}$), and $k(h_w)$ is the unsaturated hydraulic conductivity ($LT^{-1}$).

In principle, Richards' equation describes water flow due to gravity and capillary actions in the unsaturated zone [165]. The Richards equation is challenging due to its non-linearity; thus, it cannot be solved analytically [26,163]. The solutions can only be derived for specific initial conditions, boundary conditions, and hydraulic properties, thus requiring the numerical approach [166]. Numerical solutions such as the finite difference, finite element, and boundary element are generally utilized to solve Richards' equation [165,167]. Several simulation models based on Richards' equation vary from simulating soil water infiltration [163,168] to root water uptake [169,170].

Among the developed numerical packages, HYDRUS, which is based on the finite element method, is the most popular package to solve not only Richards' equation but also the convection–dispersion equation [171]. Therefore, HYDRUS can simulate the movement of water, heat, and dissolved substances in the saturated media at the one-, two-, or three-dimensional level on a small scale [172]. Other advantages of HYDRUS are having a comprehensive additional module for different applications, an intuitive graphical user interface, and detailed documentation [173].

Besides HYDRUS, some numerical packages have been developed, including, but not limited to, CATHY [174], FEFLOW [175], TOUGH [176], VS2DI [177], SWAP [178], and RZWQM2 [179]. Some numerical packages are designed for a particular purpose. For instance, FEFLOW is built for groundwater simulation, where Richards' equation is used to solve the variable flow in an unconfined aquifer [173]. The applications of SWAP and RZWQM2 are related to agricultural management, such as soil water and crop-related processes, and tend to use the one-dimensional Richards equation [173]. For example, Ma et al. [180] utilized SWAP to simulate the impact of deficit irrigation strategies on the field water balance to obtain the optimum irrigation schedule. Other features of SWAP are the simulation of crop growth, transpiration, soil evaporation, the interaction between soil moisture and surface water management, and nutrient transport [178]. Similar features are also found in RZWQM2, which was developed to model the agricultural system's physical, chemical, and biological processes.

### 5.2. Convection–Dispersion Equation for Solute Transport

The movement of contaminants from the vadose zone into the aquifer is one of the processes requiring significant attention. These contaminants might have different sources and chemistries. For instance, in agricultural areas, the source of aquifer contaminant could come from nutrients through the leaching process. In the context of sustainable agricultural management, measurement of a nutrient cycle (the amount of nutrient to enter, transport within, and leave soil) is necessary as it will affect the plant growth and groundwater

quality [26]. Hence, soil modeling to predict solute movement in the vadose zone at various spatial and temporal scales is part of precision agriculture management.

Essentially, the mechanism of solute transport involves convection, diffusion, and dispersion [181]. Considering these basic mechanisms, the transport and leaching of a dissolved substance within the vadose zone can be expressed by the convection–dispersion equation [26] as described below:

$$\frac{\partial \theta Rc}{\partial t} = \frac{\partial}{\partial z}\left[\theta D \frac{\partial c}{\partial z} - qc\right] - \varnothing \tag{2}$$

where $R$ is the retardation factor of adsorption (-), $c$ is the concentration of the solution ($ML^{-3}$), $D$ is the coefficient of dispersion ($L^2T^{-1}$), $q$ is the volumetric fluid flux density ($LT^{-1}$), and $\varnothing$ is the sink or source of solutes ($ML^{-3}T^{-1}$). By incorporating provisions for nonlinear nonequilibrium reactions between solid and liquid phases, and linear equilibrium between liquid and gaseous phases, adsorbed and volatile solutes in the vadose zone, such as pesticides, can be measured and evaluated.

The convection–dispersion equation has been applied to simulate solute transport [182,183] and salt movement or concentration, resulting in soil salinization [184,185]. Most of these simulations are performed using HYDRUS. Besides HYDRUS, other numerical packages also have been utilized, such as SWAP [178], COUP [186], MACRO [187], DAISY [188], and TOUGH [176]. SWAP and DAISY are primarily used to support agricultural management. COUP was originally developed for forest soil and lately has been modified for soil with other canopies; MACRO is used to assess the contaminant transport in structured soil, while TOUGH focuses on hydrogeological features that could be useful to assess complex processes in the near surface or deeper unsaturated zone [161].

The convection–dispersion equation poses several challenges, one of which is the simplification of the model [189]. For example, this equation considers that the dispersion coefficient and convection velocity will remain constant with depth and time, despite the field or laboratory experiment showing the opposite [189,190]. The numerical solution used for the convection–dispersion equation must be accurate since the time scale of the transport process is large [191]. The Eulerian, Lagrangian, and combined Lagrangian–Eulerian methods have been seen as viable numerical solutions [26,191].

**Table 3.** Different applications of numerical models that solve the Richards and convection–dispersion equations.

| Model | Applications | References |
|-------|-------------|-----------|
| HYDRUS | Simulation of soil water and salt transport | [192] |
| SWAP | Evaluating irrigation practices and determining the optimum irrigation schedule | [180] |
| DAISY | Predicting nitrogen leaching in cultivated area | [193] |
| COUP | Simulating the effect of low soil temperature on transpiration | [194] |
| MACRO | Modeling chlorpyrifos transport in agricultural field | [195] |
| FEFLOW | Simulating transport of contaminant in synthetic aquifer | [196] |
| RZWQM2 | Modeling phosphorus dynamic in agricultural field | [197] |

## 6. Discussions

Maintaining and improving soil quality is essential to secure food production and can be achieved by the regular monitoring of soil properties. Therefore, accurate measurements of soil properties are considered as a preliminary step for the successful implementation of precision agriculture as their variabilities could affect a variation in crop yield. In addition, understanding their variabilities in space and time will be essential and crucial for the

decision-making process in agriculture management. The agronomic soil properties required to be observed regularly include supporting factors such as SM, SOM, soil nutrients, soil texture, soil structure, and degrading factors such as soil compaction and salinity. Conventional soil sampling and laboratory analysis are accurate but time-consuming and labor-intensive. Adequate information on the spatial distribution of soil properties can support the implementation of precision agriculture strategies. However, their acquisition could be restricted by the cost related to laboratory analysis. Remote sensing observation, geophysical surveys, and soil modeling offer alternative solutions for soil property assessment in a non-invasive and time- and cost-effective way.

The use of remote sensing observations for modern agricultural management has progressed tremendously due to its advantages in observing many parameters required for precision farming. Moreover, their spatial and temporal coverages of current and future missions have progressed significantly. The utilization of optical remote sensing in the wavelength domain of visible (VIS), near-infrared (NIR), and short-wave infrared (SWIR) in soil studies has been highlighted by numerous studies. Several physical and chemical soil attributes, including SM, SOM, soil nutrient, soil texture, and soil structure, can be successfully retrieved using these spectral regions.

Despite its advantages, the inability of the optical satellite to penetrate cloud and vegetation cover has been recognized as a major limitation. On the other hand, microwave-based remote sensing can measure soil properties under a variety of topographic and vegetation cover without being constrained by weather conditions. The most popular application of microwave sensing in soil studies is SM monitoring due to its superiority in SM retrieval. However, the ability of microwave sensing is often limited by its temporal and spatial resolutions; thus, it cannot be well performed at a farm scale. Among the developed remote sensing technologies, thermal sensing applications are still limited. Commonly, thermal sensing is integrated with optical sensing to assess SM variabilities through the triangle method. The use of an optical and thermal camera mounted on a UAV might overcome the limitation of SM measurement through microwave sensing. Other soil properties can be also addressed without worrying about the impact of cloud cover due to the UAV's low altitude.

Proximal sensing (geophysical acquisition) offers the opportunity to bridge the gap between small-scale point-based measurements and large-scale remote sensing observations. Geophysical techniques can help to understand the underlying processes regulating the soil–plant–atmosphere continuum. Their techniques in precision agriculture include ER, EMI, GPR, and seismic, which differ in the type of measured physical properties. Similar to remote sensing, various soil properties have been successfully monitored using geophysics, such as SM, SOM, soil structure, soil texture, soil compaction, and soil salinity. Among them, the application of seismic methods in agricultural studies is still limited to soil compaction monitoring. Geophysical techniques have different sensitivity to soil properties. For instance, GPR and seismic methods respond primarily to the soil interface, while ER and EMI respond to bulk properties [128]. Therefore, the combination with geophysical measurement might become a future trend in agricultural applications to better understand subsoil characteristics.

Numerical soil modeling generally exploits the Richards equation and the convection–dispersion equation to simulate hydrological fluxes and dissolved substance transport in the vadose zone, respectively. Their implementations would be beneficial in understanding the physicochemical processes, particularly in the complex agricultural system. Various numerical packages have been developed to solve both the Richards and convection–dispersion equations, including HYDRUS, CATHY, FEFLOW, TOUGH, and VS2DI. Among them, HYDRUS is the most widely used numerical software to simulate underlying processes such as infiltration, root water uptake, soil contaminants, and the salinization process. Some limitations appear since Richards' equation is basically a single-phase flow equation where the contribution of air flow in the soil is considered not significant [173].

Other challenges in soil modeling could come from the complexity of physical and biochemical processes in the unsaturated zone, the location of the targeted simulation area at the interface between different spheres, and computational difficulties in dealing with highly nonlinear and coupled processes [176]. Therefore, future work should incorporate various physical, chemical, and biological parameters into models. Since vadose zone processes are part of a large environment, coupling a small-scale with into a large-scale model is necessary for a better understanding of complex natural processes—for instance, coupling between HYDRUS and groundwater models such as MODFLOW. Lastly, developing advanced numerical and visualization code sets with a friendly user interface will be a future challenge that should be addressed.

Overall, remote sensing techniques offer the ability to image the Earth's surface at a local, regional, or even global scale through different platforms, from airborne to satellites. They have been proven as reliable tools to acquire soil properties on the soil surface and below the soil surface at up to a few centimeters. Meanwhile, near-surface geophysics are usually deployed to characterize detailed subsoil properties due to their abilities to perform deeper acquisition. In particular, information derived from geophysical surveys can be used to validate remote sensing observations and soil modeling. Soil modeling is commonly utilized to predict the complex natural processes occurring in the vadose zone. Input for soil modeling can be from remote sensing and near-surface geophysics. The comparison of these three different techniques is shown in Table 4.

**Table 4.** Comparison of remote sensing, near-surface geophysics, and soil modeling.

| Remote Sensing | Near-Surface Geophysics | Soil Modeling |
|---|---|---|
| Providing lateral distribution of soil surface information at field, regional, or global scale. Resolution is too coarse for field-scale applications | Providing detailed vertical and lateral distribution of soil information in the vadose zone, generally at field scale | Providing the prediction of complex physicochemical processes in the vadose zone. Preferable to be performed at smaller domain or scale |
| Serving as initial survey to show how soil properties vary over the field and determining the grid of geophysical acquisition | Bridging the gap between remote sensing with point measurement. Can perform to validate results derived from remote sensing monitoring and soil modeling | Predicting the change in physicochemical processes over short or long periods of time in the future |

## 7. Conclusions

This study highlights various techniques and their applications in the agriculture field. Overall, remote sensing, geophysics, and modeling are promising tools to retrieve agronomic soil properties in order to implement precision agriculture. The spectroscopy method has several potential uses in soil property assessment, such as SOM, soil texture, and soil nutrients, while the microwave sensor is known for its superiority for SM monitoring. Thermal sensing is usually employed along with optical imagery to quantify SM at the field scale. All these approaches can provide lateral or spatial variations in soil properties within the farm. On the other hand, geophysics offers detailed vertical resolutions of soil properties. ER and GPR are widely used for soil moisture monitoring, while EMI is regularly applied to map soil salinization. In particular, the extension of ER, which is ERT, has been utilized recently to understand the root water uptake process in the root zone. In addition, the seismic method offers the opportunity to map soil compaction. Remote sensing and geophysics complement each other and are considered essential tools for the future to support sustainable agriculture management. Moreover, numerical soil modeling can describe numerous complex processes in the vadose zone to evaluate irrigation practices and improve agricultural management.

**Author Contributions:** Conceptualization, P.S.; formal analysis, A.P., P.S. and N.K.; writing—original draft preparation, A.P., P.S., N.K., M.D., Z.D., M.M., M.A. and P.K.; writing—review and editing, A.P., P.S., N.K., M.D., Z.D., M.M., M.A., B.T. and P.K.; supervision, P.S., N.K., M.A. and B.T.; project administration, P.S. and M.Y. All authors have read and agreed to the published version of the manuscript.

**Funding:** This research received no external funding.

**Institutional Review Board Statement:** Not applicable.

**Informed Consent Statement:** Not applicable.

**Data Availability Statement:** Not applicable.

**Acknowledgments:** The authors would like to acknowledge the support provided by the College of Petroleum Engineering and Geosciences and the Interdisciplinary Research Center for Membranes and Water Security, King Fahd University of Petroleum and Minerals, Saudi Arabia.

**Conflicts of Interest:** The authors declare no conflict of interest. The funders had no role in the design of the study; in the collection, analyses, or interpretation of data; in the writing of the manuscript, or in the decision to publish the results.

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
