# Peer review of "Remote Sensing, Geophysics, and Modeling to Support Precision Agriculture—Part 1: Soil Applications"

_water, doi:10.3390/w14071158_

Round 1

Reviewer 1 Report

The MS aims to provide for a proper review of the state of the art on methods and approaches to provide for soil information, aiding support to precision agriculture.

The MS is well structured and methodically well planned.

I have no futher comments.

Reviewer 2 Report

Dear Authors, Dear Editor,

I have read and evaluated the manuscript entitled "Remote Sensing, Geophysics, and Modeling to Support Precision Agriculture –Part 1: Soil Applications” by Pradipta et al that has been submitted for publication in Water.

This extensive review paper lays a very good foundation for the next review(s).  The authors have correctly identified that maintaining and improving soil quality is essential to secure future food production and that the starting point is by regular monitoring of soil properties (soil moisture, organic matter content).  To aid in the decision making for these farmers/growers a better understanding of soil properties can be obtained through the use of new precision agriculture technologies.  The paper then goes on to explain the pros and cons of using different wavelengths and proximal sensing or a combination to monitor these different soil properties.

The manuscript was received as a pdf document.  Overall the paper is very well written, however a few minor edits are suggested, as follows:

Line 24, ‘dan’ should be changed to ‘and’.

Line 57, change ‘those’ to ‘these’.

Line 96, change ‘plays’ to ‘interacts’.

Line 99, change ‘high’ to ‘higher’.

Line 123/124, change ‘make efficiency and accuracy of nutrient prediction become a key factor’ to ‘make prediction of nutrient efficiency and accuracy a key factor’.

Line 142/143, change ‘The latter role makes SM’ change to ‘The latter role on SM’.

Line 188, change ‘The use of a machine learning algorithms’ to ‘The use of machine learning algorithms’.

Line 205/206, change ‘agriculture soil that is’ to ‘agricultural soils that are”. thus 253

Line 254, ‘allowing to measure SOM through the same algorithm’ change to ‘allowing SOM to be measured through the same algorithm’.

Comments:

There needs to be consistency to the references.  Three different styles are used in the review, for example [27], [27], 27.  

This review has been extensively referenced with 195 references, the authors are to be congratulated.

Overall, this review evaluates numerous methods of monitoring important soil properties that influence leaching and surface runoff.  The precision agriculture technologies described offer opportunities for surface, sub-surface and vadose zone monitoring and modelling.  

Therefore, it is my recommendation that this paper should be included in the upcoming issue of Water.

Reviewer 3 Report

The authors summarized Remote Sensing, Geophysics, and Modeling methods used in quantifying soil properties. The topic is interesting and falls in the scope of the journal. The manuscript can be considered for possible publication after major revision.

(1) Abstract: The background should be concise, and major points of the review and implications from the literature review should be added.

(2) Detailed comparisons of different methods are preferred.

Reviewer 4 Report

This manuscript describes very superficially the various non-invasive techniques such as remote sensing, geophysics, and soil modelling have been successfully employed in agricultural studies by a number of studies, in order to characterise the soil of a farm, and thus plan the most appropriate fertilisation and irrigation for the crops. The review of the work published so far has been very extensive and up to date. However, the description of the methodology is very generic and does not provide guidelines that could help the reader to choose the most suitable one for their specific conditions. For example, spectral reflectance values for each type of component. Perhaps it would be interesting to describe in more depth the works cited by the authors. Some comments I have attached to the manuscript.

Round 2

Reviewer 3 Report

The authors have revised the manuscript following reviewers' comments.

Reviewer 4 Report

The authors have improved the first version of the manuscript by incorporating more specific data relating to the different technologies described. The manuscript thus acquires the minimum value necessary to be referenced in future work on this subject.